# “Living with a Stoma”: Exploring the Lived Experience of Patients with Permanent Colostomy

**DOI:** 10.3390/ijerph18168512

**Published:** 2021-08-12

**Authors:** Areti Stavropoulou, Dimitrios Vlamakis, Evridiki Kaba, Ioannis Kalemikerakis, Maria Polikandrioti, Georgia Fasoi, Georgios Vasilopoulos, Martha Kelesi

**Affiliations:** Department of Nursing, School of Health and Care Sciences, University of West Attica, 12243 Athens, Greece; astavropoulou@uniwa.gr (A.S.); vlamakisdimitris@gmail.com (D.V.); ekaba@uniwa.gr (E.K.); ikalemik@uniwa.gr (I.K.); mpolyk@uniwa.gr (M.P.); gfasoi@uniwa.gr (G.F.); mkel@uniwa.gr (M.K.)

**Keywords:** colostomy, assessment, patient outcomes, quality of life, nursing care, qualitative research

## Abstract

Introduction: Living with a permanent colostomy brings severe changes in patients’ lives. The general health status as well as the personal, social and professional life of patients are significantly affected. Aim: The aim of the present study was to investigate the lived experience of patients undergoing permanent colostomy. Material and Methods: A qualitative research design based on interpretive phenomenology was carried out. Semi-structured interviews were conducted as the data collection method to obtain in-depth information regarding the research topic. The study sample consisted of eight (8) patients who had undergone a permanent colostomy. The data analysis was performed by the method of content analysis. Results: From the analysis of the data, three main themes emerged, namely: (A) Experiencing a traumatic event; (B) Living a new reality; (C) Efforts to improve quality of life. Five subthemes were formulated which were encompassed within the respective main themes accordingly. Conclusion: Patients with permanent colostomy face significant life changes that are experienced in a traumatic way. Issues such as autonomy, family and organizational support, self-management and empowerment can significantly improve the patients’ quality of life. Further research, regarding caregivers’ experience, improved community nursing care as well as nurses’ views on the needs of colostomy patients and their families, is suggested.

## 1. Introduction

Colostomy creation is one of the most common therapeutic interventions applied to pathological conditions of the colon and mainly to colorectal cancer [1]. Although colostomy is considered the most suitable therapeutic approach, it can inevitably cause problems for both physical and psychosocial functioning and affect quality of life (QOL) by altering the patient’s self-image, defecation patterns and lifestyles [2].

Living with a colostomy negatively affects overall QOL. The presence of colostomy is associated with a variety of complex problems, such as psychological and sexual problems, dissatisfaction with altered body image, changes in daily routine, travel difficulties and tiredness. Factors such as age, gender and time since treatment may have an effect on ostomy-specific QOL; however, relevant literature does not provide conclusive evidence about this effect [1]. Research demonstrates issues of uncertainty, anxiety personal, professional and social life limitations and a tendency to loneliness, that may lead the person towards complete social isolation [3,4,5]. The adjustments that occur in a person’s life with colostomy significantly affect his interpersonal and intimate relationships. The adoption of a new life model adapted to the colostomy management has an emotional impact on the patient himself and the people in his close environment, such as his family and sexual partner [6]. Married individuals experience major sexual life changes, especially right after surgery or at the early stages of the disease. These changes however are gradually resolved as soon as patients learn to effectively handle the new situation [3]. Furthermore, women appear to have a lower QOL compared to men and this relates to body image changes, fears of rejection and an inability to carry out household tasks [6]. The lives of stoma patients also seem to be affected by aging, as elderly people experience more constraints in physical functions, while young people are more concerned with emotional issues and future economic perspectives. Family income may also have an impact on the lives of patients due to the high cost of the materials needed for ostomy care [6]. The professional lives of colostomy patients are severely affected, as having a colostomy may provoke a partial or complete loss of work [7]. As a result, socio-economic implications arise in an individual’s life accompanied by feelings of worthlessness and incapability. Patients with permanent colostomy confront a variety of changes which lead to a reduction in social participation, as they are often isolated from others. Additionally, patients often feel that people who used to be close to them have become distant [8].

### 1.1. Patients’ Experiences with Colostomy

Patients’ experiences undergoing colostomy have been studied before and after ostomy creation. According to a relevant study, patients shortly after ostomy surgery expressed feelings of uncertainty and anxiety derived from the disease and the ostomy itself, and these negative emotions remain for quite some time after surgery [7]. Another study [9], reported significant changes that patients experience on a psychological and emotional level. This study focused on patients’ living experiences with colostomy and concluded that the participants in the study considered colostomy to be so traumatic that even death is thought to be another alternative option [9].

Research evidence [7] also referred to patients’ feelings of identity transgression, changes in quality of life, roles and relationships, embarrassment because of the changes in body image, fear and situations of shame and frustration. Further research [10], focused on the issue of self-esteem, indicating that colostomy patients live through a powerful experience, causing major instabilities in their daily life.

However, colostomy creation is associated with the development of strong defense mechanisms and new perspectives for patients. Patients with a stoma, after surgery, seemed to appreciate other aspects of life which had not been considered before [7]. Experiences reported by patients were associated with relief from solving their health problems, mobilization for maximizing autonomy, self-care and management of the new situation [7]. Colostomy creation was perceived as an opportunity to continue life and strengthen companionship [11]. Relevant research highlighted that the existence of colostomy is associated with the formation of a new way of life, due to noticeable changes in patients’ daily activities [6].

In Greece, there are not many research studies on the influence of colostomy on patients’ lives. Relevant literature mainly focused on colostomy patients’ quality of life which is significantly affected, leading to psychological, sexual disorders and social isolation [12,13]. In addition, there are no qualitative studies that explore the experiences of colostomy patients in Greece, or the consequences of colostomy on their daily lives. Therefore, a qualitative study focusing on stoma patients’ experiences and on the effects that a permanent colostomy has on various aspects of patients’ lives is considered significant.

### 1.2. Aim

The aim of the present study was to examine the lived experience of patients who have undergone a permanent colostomy, using a qualitative research approach. The findings of this study will contribute to the development of knowledge in this field by offering unique evidence based on patients’ views and real-life experiences.

## 2. Materials and Methods

A qualitative research approach was used to conduct this study. Qualitative methodological approaches are appropriate when the researcher aims to understand and extract meanings for the phenomena through the exploration of participants’ views and experiences [14]. Since the purpose of the present study was to investigate the lived experience of the patients undergoing permanent colostomy in depth—to understand how people with colostomy live with this and construct meaning in their lives—a qualitative approach based on the principles of interpretive phenomenology (IP) was followed. The aim of phenomenological research is to answer research questions by exploring and accurately describing the individuals’ lived experiences and to further understand how this experience gives meaning to people’s lives [15]. To a further extent, IP aims to determine the human existence through the way in which individuals experience and interpret their life and their world [16,17].

### 2.1. Sampling Strategy

Purposeful and snowball sampling strategies were used in the present study. Purposeful sampling is the intentional selection of participants that, according to the personal judgment of the researcher, are considered as the most representative of the population under study. Snowball sampling enables the researcher to access and identify potential participants suggested by other study participants in the study [16]. These strategies are mainly used in qualitative research when the researcher wants to identify and recruit people with specific characteristics who may offer wealth of information for the topic under investigation, known as information-rich cases [18].

### 2.2. Study Population-Sample

Patients with permanent colostomy consisted the study population. Inclusion criteria were formed as follows:Adults who have undergone a surgical creation of a permanent colostomyPatients who lived with a permanent colostomy for at least one yearParticipants who were not hospitalized during the study.

The above criteria were considered necessary for the participants to provide rich information, which gained through their valuable experience from living with the colostomy. 

The study was contacted in the region of Attica, which is one of the 13 regions of Greece, based in Athens, and is divided in eight peripheral units, including more than 40 municipalities. Potential participants were mainly identified through the researchers’ personal contact with the community nurses who provided home care to patients with permanent colostomy in three different peripheral units. The researchers informed the community nurses about the commencement of the study and asked them to advertise it to the patients who had a permanent colostomy, requesting their participation in the study. After that, the researchers contacted the patients who were interested in participating in the study and informed them in detail about the purpose of the research, and the data collection techniques. The potential participants were also fully informed about the voluntary nature of the study. Issues of confidentiality and protection of personal data were emphasized. Finally, the informed consent form was signed by the participants before data collection. Participants from 3 different peripheral units of the region of Attica were included in the study. This distribution was considered essential for obtaining a breadth and wealth of information from different settings and from participants with sufficient knowledge and experience of the phenomenon under study.

### 2.3. Data Collection

Data collection was performed through semi-structured interviews. The use of semi-structured interviewing is perhaps the most common method of data collection in qualitative research [19]. Through semi-structured interviewing, the researcher is able to collect rich data by giving the participants the opportunity to express themselves freely [14].

The semi-structured interview-guide included one open-ended question that invited participants to describe their life experiences after the colostomy. Some additional questions, which concerned the participants’ daily activities after the colostomy, were also asked only when further facilitation of the discussion was considered necessary. Interview questions were formed as follows: 

**Introductory** **Question:**
*Please describe your life experience from the moment you experienced the colostomy onwards.*
**Additional** **questions** **(to** **be** **used** **if** **necessary):**
*How has your daily life been after the colostomy?*

*Tell me about the changes, the challenges and the difficulties you faced after the colostomy.*

*How could your life be improved?*


The interviews were conducted at a place and time chosen by the participants. All the participants interviewed in their home, as this location was considered to be more convenient for them. Privacy and confidentiality of data were ensured throughout the interview process. A tape recorder was used during the data collection phase. Prior to the use of the tape recorder, the permission of the participants was requested. The interviews lasted about 20 to 35 min and were completed when the participant had nothing else to add to his/her narrative. The data were transcribed verbatim and converted into written text upon the completion of the interview. All interviews were conducted in Greek. The participants’ passages were translated into English and accuracy of the translation was ensured by using a backward translation technique [14]. Each interview was assigned a code that was used throughout the analysis of the data in order to fully preserve the anonymity of the participants. The data collection phase was completed when data saturation occurred [20]. In the present study, data saturation was reached when the eight interviews were completed, and the collection of new qualitative data did not make a significant additional contribution to our study findings.

### 2.4. Data Analysis

Data analysis was conducted based on the principles of IP and focused on demonstrating the participants’ experiences and identifying themes and patterns reflected in participants’ own words [21,22].

The data analysis in the present study was performed manually, following the three basic steps which described by Pietkiewicz and Smith [23], and include: (a) detailed, multiple reading of the data and formulation of notes; (b) converting notes into emerging topics; and (c) searching for relationships between topics and grouping. Based on these areas, the analysis of the data leads to the representation of the meaning of an experience based on the recognition of important themes and common patterns (themes and patterns), and is captured in the words of the respondents themselves [14,15].

Self-reflective notes were kept by the researcher throughout the study. That was an important part of the data analysis and assisted the researchers to gain an in-depth understanding of the participants’ experiences.

### 2.5. Ethics

The study participants were informed in detail about the purpose and nature of the research. Voluntary participation in the study was emphasized and the participants’ rights to autonomy and privacy were preserved throughout the research course. Participants were asked to speak freely about their experience and life with the colostomy. They were also informed that they had the right to withdraw from the study at any time with no impact on their care. Special mention was made on issues of managing data confidentiality and anonymity before, during and after the interview. The researcher informed the participants that only the researchers will have access to the research data, which will be used for scientific purposes only. It was mentioned that interview data will be stored for a reasonable period of time and then will be destroyed. Full access to the results of the study was also provided to the participants upon their demand. Consent was obtained by all participants in this study prior to the data collection phase. 

Thesis Review Committee of the Postgraduate Program “Wound Care and Treatment” of the Department of Nursing of the University of West Attica issued approval for this research under REG NUMBER 9/9-10-2019/6. All procedures performed in the studies involving human participants were in accordance with the ethical standards of the institutional and/or national research committee and with the 1964 Helsinki declaration and its later amendments or comparable ethical standards.

### 2.6. Trustworthiness of the Research

Strategies for ensuring the trustworthiness of qualitative studies include prolonged engagement, persistent observation, triangulation, member checks, thick description audit trails, researcher’s diaries and reflexivity [24,25]. In the present research, trustworthiness was ensured through the use of reflexivity and triangulation strategies. In particular, with regard to reflexivity, a diary was kept by the researcher throughout the study which allowed the researcher to control personal values, perceptions and prejudices. Regarding the triangulation strategy, the analyst–triangulation method was used. Two researchers analyzed and interpreted the data so the cross-referencing of the relevant data could lead to reliable results. Using this technique, the researcher’s possible biases in data interpretation were reduced. The emerging themes and subthemes of the study were confirmed after repeated and careful reading of the data by the two researchers and the presentation of the data in a meaningful and reliable way was ensured [26]. 

## 3. Results

Eight (8) individuals with a permanent colostomy, aged 48 to 77 years, participated in the present study. All participants were married and were living with a permanent colostomy for at least one (1) year. The participants’ characteristics are presented in Table 1.

The analysis of the data revealed three (3) main themes, namely: (A) Experiencing a traumatic event; (B) Living a new reality; and (C) Efforts to improve quality of life. Five (5) subthemes emerged: (A1) Extreme emotions; (B1) Personal changes; (B2) Social adjustments; (C1) Autonomy; and (C2) Support. Main themes and subthemes with representative quotes are presented in Table 2.

### 3.1. Main Theme A: Experiencing a Traumatic Event

Participants were initially asked to describe their experience from living with a permanent colostomy. The participants displayed an array of negative feelings that they said dominated their life before and after the surgery. A sub-theme called Extreme Emotions was formed that involved the participants’ emotional responses to living with a colostomy. 

#### Extreme Emotions

Extreme, dominant emotions such as fear, repulsion, self-deprecation and shame were expressed by the participants. Anxiety, uncertainty and negative feelings from the altered body image after the creation of the colostomy were described in a dramatic way.

“*I felt fear and agony with the thought of my new body image… I felt, let’s say, hurt by it … It was like going to war, having been injured and getting a permanent scar that reminds you of that moment, every day*” (p. 8).

“*I was very anxious about the surgery and how I would be after that. I could not look at the stoma… I was scared and shocked to see something like that on my body…*” (p. 7).

While describing their experience, the participants conveyed feelings of an inability to understand the new situation which occurred after the creation of the stoma. In addition the fear of death was communicated explicitly, a fear which was often linked to the family’s future life and prosperity.

“*It was a tragic experience, it happened so fast that I did not understand what exactly was going on*” (p. 4).

“*I was afraid of dying and how they (the patient’s family) will manage it*” (p. 4).

The stoma creation seemed to be a traumatic event not only for the patients but also for their significant others, especially their partners. This appear to affect intimate relationships and sexual life. 

“*My partner was afraid of the stoma…afraid to touch me at night in case this might hurt me… and I am embarrassed to have the bag and to smell it when we are lying down… I do not want them to see me naked…*” (p. 1).

Anger, denial and sadness were often the response to the traumatic event of getting a permanent colostomy. The loss of life control was also experienced intensely by the participants as they felt that their previous life routine had been completely changed.

“*In the beginning I was very angry… I could not control myself… I was shouting at my wife without any reason… I was thinking why did this happen to me…*” (p. 3).

“*It was very difficult for me…I have not accepted it yet…but now nothing can turn back time, nothing can change…*” (p. 7).

### 3.2. Main Theme B: Living a New Reality

Participants were further discussed their daily lives after the creation of the colostomy. Notions of struggling and rigorous scheduling of their personal, professional and social lives were revealed. Two sub-themes were formed within this main theme called Personal Changes and Social Adjustments.

#### 3.2.1. Personal Changes

The participants referred to a variety of personal changes that the colostomy brought to their lives. Changes seemed to be inevitable in many aspects of their personal lives. The physical and psychological condition of patients seemed to be affected, making them vulnerable and dependent on others. 

“*I could not change it (the colostomy bag) myself, I was dependent on the nurse*” (p. 2).

The massive changes that individuals experienced seemed to lead to isolation, introversion and restrictions in personal life. Self-changes and loss of privacy were also highlighted by the individuals with a permanent colostomy.

“*After I left the hospital I wanted to be alone. I kept saying that my life was over. I avoided meeting with people, I didn’t want to talk…*” (p. 7).

“*I just remember the woman I was … and I can see the changes now… my life has changed… I had to learn how to change the bag, I had to let my relatives watch me to do that in case I need their help in the future ...*” (p. 5).

The creation of the colostomy affected the individuals’ lifestyle and their daily functions. Healthy sleep and bedtime routine seemed to be significantly disturbed as the participants experienced major problems in their sleep routine. 

“*I had difficulties in sleeping at night, I could not sleep, I was afraid that the bag would come off, I got up in the morning feeling tired…*” (p. 2).

Participants referred to the difficulties encountered in everyday life and the efforts made to adapt into a changing reality. Patients with colostomy followed a certain diet and activity program for maintaining a good health status and physical functions. They seemed very concerned about how to control their bowel functions and how to avoid embarrassment during their personal interactions. Anxiety and fear for the colostomy functions seemed to overwhelm the patients’ daily schedule.

“*I always plan to have the bag with me in case a change is needed and to be cautious with food to avoid any accidents*” (p. 6).

“*My daily schedule is always planned very carefully…the fear of dropping the bag is always there…*” (p. 1).

#### 3.2.2. Social Adjustments

Participants’ social lives were significantly affected after the creation of the colostomy. As the recovery can be time consuming and painful, the individuals reflected the difficulties of social adjustment.

“*When I had to go away from home… I was frightened that if something would happen, if this (the colostomy bag) dropped, who is going to help me…*” (p. 2).

“*Socializing and going out was restricted after the colostomy*” (p. 5).

The possible release of odors, leaks and intestinal noises created additional restrictions in the participants’ social lives. Most of the participants seemed to avoid interaction with others, as they described it as an uncomfortable situation.

“*Going out is a problem that requires a solution, for example it is difficult to go for long holidays… it is not comfortable going out for dinner to relatives and friends… or going to a party… and if a sound or smell comes out, oh…, I feel so much shame…*” (p. 3).

Furthermore, social isolation and protection of private lives was mentioned by some participants as a personal choice in order to avoid unpleasant behaviors.

“*Sometimes people feel sorry for you… I did not want anyone to pity me… so I prefer to be alone, not to let the others know what I am doing or that I am not who I used to be. It took me a long time to go out for a walk...*” (p. 7).

Social life was considered a key factor for adaptation and continuation of life. Continuation of professional life seemed to have an important role in the socialization of patients with colostomy.

“*Despite the fact that the adjustment to professional life was gradual and difficult at the beginning, it helped me a lot to meet my colleagues and to be together with people … to see that there was an understanding…*” (p. 4).

Participants’ narratives also showed that factors such as resilience, strength, optimism, absence of complications, fast recovery and good health act beneficially for adaptation and the continuation of social and professional lives.

“*Trying to see what is happening in a positive way, is good, no matter how difficult it is. I do not want to give up…I want to continue my daily routine and my social life…to go out with friends and have fun*” (p. 5).

The importance of a social support network including friends and family was also referred to as a motivation for the continuation of life. Age, lifestyle, living conditions and the existence of co-morbidities seem also to relate to social adjustment.

“*My friends although they were scared the first time they heard it, they then gave me a big hug to make me feel protected and that was a very strong motivation for life in general*” (p. 8).

“*… I am strong and adapt easily… or let us say that I see the colostomy as a minor problem compared to the other health problems I have…*” (p. 5).

### 3.3. Main Theme C: Efforts to Improve Quality of Life

Participants referred to the quality of life after the colostomy. The concepts of personal effort and education prevailed in the participants’ narratives. Two subthemes were formed in this section involving notions related to life enhancement. These subthemes were named Autonomy and Support.

#### 3.3.1. Autonomy

The acquisition of autonomy was reported as a major issue in improving the quality of participants’ lives after colostomy. Self-management was highlighted as a major step towards autonomy.

“*I would feel much better if I could change it (the colostomy bag) myself, if it was a little easier for me… if I could go out and go on long holidays…*” (p. 3).

From the descriptions of the participants, it seems that the acquisition of autonomy does not occur automatically. It was the result of a slow, painful process that requires personal effort and strength of character.

“*It took me many days to look at it (the stoma) … I needed to know though what to do with the bags and all these…I didn’t want others to do it for me… and I knew that nothing will change now. I just have to make my own decisions and continue…*” (p. 7).

The daily care of the stoma was a challenge that participants had to face and resolve. Provision of information and training appeared to facilitate the acquisition of autonomy. Time, experience and successful management of the colostomy by the patients themselves seemed to also support the individuals’ autonomy.

“*Learning was the key, and this helped me to start again, to succeed … to enter slowly into my everyday life routine*” (p. 1).

“*After 3 years of living with the stoma, I have overcome the obstacles… I know how to manage it now, I control myself… now it is fine for me…*” (p. 4).

Autonomy was referred to as a continuous effort that involves overcoming of many difficulties and fears for the patient and the family. Patients’ self-belief, self-knowledge and persistence in achieving autonomy and self-management emerge through the descriptions of the participants. They seemed to draw significant strength from their family environment, which contributes in various ways to the autonomy of patients.

“*After all these challenges, I believe that I gradually began to follow a fine program and I keep trying to improve my life. The children learned to live with the new reality, but always believed in me and to some extent were frightened. They help me a lot, they take care of me but they also let me have the autonomy I want. My family is my strength*” (p. 8).

#### 3.3.2. Support

Support from the State and the National Health System in terms of continuity of care and patient education, was mentioned by the participants as a factor that may improve quality of life and care after discharge.

“*Having an organized health care system that can provide the care needed after discharge is vital … it shouldn’t be the patient’s responsibility to find a person to change it (the colostomy bag) after discharge*” (p. 1).

“*A more organized form of education in terms of changing the bag, managing the stoma, learn the products for keeping the stoma clean and healthy, might make my life better*” (p. 5).

In addition, support from community health care services appears to have a beneficial effect on the patients’ life and reduce feelings of anxiety and uncertainty. The role of the community nurse is crucial, as it seems to reduce people’s stress and alleviates their problems.

“*She (the community nurse) makes life easier for me…the nurse makes the changes so I plan my work effectively, I feel clean and confident …*” (p. 3).

Moreover, issues such as staff shortages and limited staff availability may lead to fragmented care at a community setting, and this was revealed in the participants’ narratives as a traumatic event.

“*I remember the time the bag came out… it was a few days since I came back from the hospital, and I did not know what to do, none of us could to put it in the right place… I had to search and call for a private health service to do it*” (p. 5).

Education, knowledge, a supportive environment and willingness to live life were also reported by the participants as important factors that helped to improve their quality of life.

“*It is disturbing to constantly think about whether you are doing something right or wrong… I want to know how I will do it (the change) …*” (p. 7).

Participants recognized that colostomy brought about several changes in their daily lives. A willingness to live and support from others were mentioned as crucial factors for overcoming the difficulties and for improving life after a colostomy.

“*Everyday life has changed… I had to be educated in changing the bag, and others around me had to learn also in case I need their help… but as long as we are here alive, as long as we have the willingness to live, we have memories, and warm, supportive people next to us, then we must try to improve our lives…I want as much as possible to have a very beautiful and lively life, to have a good life and I can see that over the years I can achieve this*” (p. 8).

## 4. Discussion

The results of the study demonstrated that patients after the surgical creation of colostomy frequently experience a plethora of intense emotions that are mainly traumatic and unpleasant. The changes induced by colostomy in patients’ daily lives are so tremendous that the changes adversely affect their personal and social lives. Patients’ efforts to adapt to their new lives after colostomy are focused on ways to improve their quality of life through support from their social environment and health services. Acquisition of autonomy is closely related to the improvement of patients’ lives. Adaptation to new life conditions after colostomy is enhanced by factors such as partner acceptance, the ability to return to work with an acceptable environment and support from family, friends and health professionals. Factors such as the general health status of the patient, potential comorbidities and the course of recovery, directly affect adjustment to colostomy. Support provided by health care services, including the nurses’ roles in patients’ education, is determinant for the smooth transition through all the disease stages to self-management and autonomy.

### 4.1. Living with a Traumatic Experience—Extreme Emotions

The results of this study are in line with other relevant studies, which recognize the life-saving operation of colostomy as a traumatic process [3,27]. In the literature, the recovery of patients with colostomy is cited as a painful process that demands significant psychological effort and support until the patient adapts to their new daily life. Typically, colostomy is acknowledged as akin to a wound in a war that leaves indelible memories of the pain and the surgery [6]. In addition, factors and measures capable of mitigating the extent of the traumatic effect of the colostomy are reported. An important factor is the effective communication and provision of elaborate information about the type of surgery and the forthcoming changes in a patient’s life [7,28].

Capilla–Diaz et al. [7], illustrated that communication, life experiences, comorbidities and self-knowledge are factors that determine the way that patients respond to ostomy and adapt to their new everyday life. For the eight participants in the present study, each aspect in their lives was affected, inverted or reassessed. Some of these manifestations were body image, psychological state, sexuality, dynamics of the relations with family, relatives and work environment as well as changes to their economic and social lives. Similar results are reported in the studies of Jayarajah [29], and Jenks et al. [30], who showed that patients’ daily lives after colostomy involved significant changes in physical, psychological, interpersonal and professional dimensions.

### 4.2. Living a New Everyday Life—Personal Changes and Social Adjustment

Changes in body image and the possible loss of masculinity or femininity associated with patients’ relationships, sexuality and socialization are significant personal changes, according to relevant research [9,31,32]. In the present study, body change was one of the most frequently repeated difficulties experienced by participants. As shown in the literature, this change is experienced more intensely by women with colostomy and is accompanied by a reduction in social activities and a diminished quality of life [7]. The results of the present study showed that people felt vulnerable due to changes in body image for a long time after a colostomy. Similar results are reported in the studies conducted by Thorpe et al. [33], and Anaraki et al. [34], who claimed that colostomy creation is accompanied by vulnerability, emotional instability and social isolation, mainly attributed to changes in physical functioning and alterations in body image.

In addition, relevant literature demonstrated that ostomy may cause long-term or even constant absences from work, or a partial or even complete loss of work, which leads to a decline in the socio-economic status of both patients and their families as well as to negative emotions such as despair and uncertainty [7,35]. Male patients experienced role loss, as they felt they were no longer able to provide financial support to their families. Similarly, role loss was experienced by participants whose work provided them with a context to express their male identity and their role in their family. Returning to work was conceptualized by patients as an important factor for a smooth transition back to family and social life [7].

Differences were found in terms of patients’ attitudes and adaptations to their social lives. Some of the patients who desired to maintain their social network shared their health issues, whereas others preferred to conceal their health status due to fear or anxiety about others’ reactions. According to the literature, this concealment may trigger stressful and traumatic situations, leading patients towards social isolation [35,36].

Social support was a fundamental component for effective management of social difficulties. The support of family and friends alleviated the situation. Similar results are observed in relevant studies [37,38], which pointed to the family as a primary source of social support for patients with colostomy. As reported by participants, the partner plays a central role in emotional support issues including care. According to relevant research, some patients felt incompetent, nervous and aggressive when they did not enjoy the support of their family [7,39]. Sun et al. [40], acknowledged the vital role of family support to the loved person with colostomy through efforts to accept the new figure and help them recover their self-esteem. Social isolation is seen as a consequence of a lack of emotional and social support, mainly by family. Emotional support seems to be crucial for improving patients’ quality of life, with family playing the key role.

### 4.3. Efforts to Improve Quality of Life—Autonomy and Support

Everyday life, upon returning to society, involves moments of significant difficulty and complex feelings for the patient, such as shame, fear, insecurity and most importantly, embarrassment. At the initial stage, these factors seemed to have only a negative aspect. However, these traumatic experiences may help patients with ostomy to develop coping strategies and become stronger [38]. Surgery may change the life of the individual and his family but these changes also reflect his quality of life. Interestingly, only when the patient returns to the activities he used to get involved in before the surgery and comes to terms with his new self and the new reality does he begin to live better, thus understanding the potential of life offered by ostomy [41].

Nursing interventions can alleviate the adverse effects caused by ostomy and help patients to improve their quality of life [42]. Sun et al. [40], concluded that, even if a person with colostomy has attained adjustments and a high quality of life, the restoration of normal intestinal function is still desirable and expected. This finding was not apparent in the descriptions of the participants in the present study. It can only be considered as an expression of desire, especially by patients who strongly and traumatically recall the feeling of discomfort and emphasize their refusal to accept the colostomy as part of their body function from now on. Patients felt that ostomy care was a burden on their family. Conflict, denial, stress and isolation processes were described in the results of relevant studies [7,36].

The factor of autonomy seemed to provide a special meaning for patients with permanent colostomy. According to participants’ experiences, it seemed that they all achieved a significant degree of ability to maintain autonomous care or management of the ostomy, in a supportive and protective environment. According to the findings of Reinwalds et al. [43], the concept of autonomy in life with the ostomy presupposes the acceptance of uncertainty and the loss of control of bowel movements, and is characterized by insecurity. This finding, however, does not mean that patients’ ability to live with sufficient independence is suspended or excluded.

Patients and their families need to be supported and trained in ostomy management. Despite this support and training, only the necessary behavioral and environmental changes will finally lead to autonomy. The concepts of quality of life, support, education and autonomy seem to be interactive and interdependent [44,45]. Patients and family are called to be trained in the use of the ostomy mechanism and its components (bags, bases, auxiliary materials, etc.). Additionally, they are encouraged to enhance their confidence and skills in ostomy care. Indeed, living with a colostomy means experiencing various and multiple barriers towards managing effective self-care postoperatively [31]. These barriers undermine the ability to establish self-confidence around ostomy care, thus adversely affecting quality of life and patient autonomy [42]. People with colostomy can enhance their autonomy through training and the use of smartphone technology involved in ostomy care. These technologies may contribute to an improvement in autonomous living and in ostomy care while at the same time mitigating the mental and social burdens associated with ostomy [46]. Results of the above studies confirm the descriptions obtained from the participants in the present study. Issues such as the existence of a more structured form of support, provided by health services and health professionals, as well as education in terms of managing the products used in the ostomy, were mentioned by the participants in this study.

All the above support mechanisms exert a positive influence on patients’ self-management and improvement of their lives. According to Mota et al. [9], patients with ostomy develop feelings of uncertainty, fear and insecurity towards the possibility of experiencing difficulties related to the use and supply of colostomy materials and other products necessary for the proper maintenance of ostomy. Although this concern was not explicitly expressed in the present study, respondents’ desire to ensure autonomous living and care of their ostomy reinforced the need for elaborate training and support by services and health professionals.

In each country, cultural, social, economic and religious differences affect the behavior of patients with ostomy and their families. Interestingly, it was observed that in Greece, few studies have explored the cultural and social parameters towards patients with ostomy. However, it is well known that when a patient is diagnosed with cancer, there is a spirit of overprotectiveness from the part of both the physician and the family [47]. According to Vasilopoulos et al. [48], Greek patients with ileostomy experience reduced mobility and independence of living in post-surgery period that minimizes their ability to conduct self-care and limits their social life as a result of physical impairment and emotional trauma [48].

## 5. Limitations and Suggestion for Further Research

Qualitative methods, inherently, do not aim to generalize the study results but instead explore the uniqueness of a phenomenon in-depth. Therefore, findings of qualitative research cannot be extrapolated to wider populations. Results of the present study should be viewed under this constraint. Difficulties in approaching a sensitive population such as patients with permanent colostomy was the main limitation of the study. Although the information collected by the participants in the present study was considered sufficient for the purposes of the research, additional data might contribute to further advancement of knowledge in the field. Suggestions for further research include exploring the experience of caregivers and families of patients with colostomy, investigating the views of stoma patients regarding home care as well as investigating nurses’ views on patients’ and families’ needs.

## 6. Conclusions

Permanent colostomy is a painful experience, imposing restrictions on a patient’s personal and social life and affecting severely the individuals’ mental and social well-being. The current study indicates that stoma patients are frequently accompanied by negative feelings such as anger, anxiety and fear. The changes induced by colostomy in patients’ daily routine are so tremendous that they adversely affect all aspects of life, including intimate relationships, professional conditions and financial conditions. Adaptation is a long-term process that requires effective communication, acceptance and understanding. The individuals’ effort to improve their lives is constant. Emotional and organizational support, autonomy, education and self-management may improve quality of life. A nurse is the key person for assisting stoma patients to gain empowerment, support and acceptance and develop the appropriate mechanisms for managing and adapting to the new life conditions effectively. Knowledge derived from this research could positively impact both clinical and community care. The study findings indicate that the assessment and diagnosis of anxiety and fear disorders should be a priority for clinicians caring for stoma patients. Clinicians should also consider the fact that the general state of patients’ health, potential comorbidities and the course of recovery directly affect adjustments to colostomy. Taken together, these findings suggest that the diagnosis of comorbidities or disorders such as anxiety may lead to a more efficient care plan. Moreover, holistic assessment and management of stoma patients are crucial issues in community nursing care. Community nurses as interdisciplinary team members should acquire specialized knowledge, skills and training on stoma care for providing personalized and high-quality care to stoma patients. More frequent visits may also enhance community nurses’ knowledge in terms of early recognition, prevention and treatment of complications and addressing adverse events. Enhanced communication skills and patient education are essential features that may assist self-care management and patient empowerment. Development of digital skills, identifying and implementing evidence-based practice, are also areas in which community nurses need to be specialized for providing high standard care to stoma patients and their families.

## Figures and Tables

**Table 1 ijerph-18-08512-t001:** Demographic characteristics of the study participants.

Participants Code Number	Gender	Age	Educational Level	Profession	Years Living with Colostomy
P1	F	63	High School	Household	1
P2	M	77	High School	Retired	2
P3	M	57	High School	Free-lance professional	1
P4	M	48	High School	Free-lance professional	3
P5	F	52	University	Disability Retirement	4
P6	M	67	University	Retired	12
P7	F	49	High School	Household	2
P8	F	60	University	Retired	7

**Table 2 ijerph-18-08512-t002:** Main Themes and Subthemes.

Main Themes	Subthemes	Representative Quotes
A. Experiencing a traumatic event	A.1. EXTREME EMOTIONS	“*I felt fear and agony … It was like going to war, being injured and getting a permanent scar that reminds you of that moment, every day*” (p. 8).
Β. Living a new reality	Β1. PERSONAL CHANGES	“*I just remember the woman I was … and I can see the changes now… my life has changed …*” (p. 5).
	Β2. SOCIAL ADJUSTMENTS	“*Sometimes people feel sorry for you … I did not want anyone to pity me … so I prefer to be alone, not to let the others know what I am doing or that I am not who I used to be. It took me a long time to go out for a walk …*” (p. 7).
C. Efforts to improve quality of life	C1. AUTONOMY	“*After 3 years of living with the stoma, I have overcome the obstacles … I know how to manage it now, I control myself … now it is fine for me …*” (p. 4).
	C2. SUPPORT	“*She (the community nurse) makes life easier for me …the nurse makes changes so I can plan my work effectively, I feel clean and confident …*” (p. 3).

## Data Availability

Data generated during the present study cannot be shared due to of the need to preserve the subjects’ privacy and confidentiality.

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
