# Peer review of "“Living with a Stoma”: Exploring the Lived Experience of Patients with Permanent Colostomy"

_ijerph, 2021, doi:10.3390/ijerph18168512_

Round 1
Reviewer 1 Report
Dear authors,
Thank you very much for the effort in conducting this study as well as the possibility to read it.
I have few comments:
Introduction part:
- I’d like to see more information in this part regarding overall according to age, gender and other sociodemographic characteristics
Data collection:
I which town, area this study was conducted, is there any possibility for bias?
Why did you decide to interview 8 patients? Where this interviews were conducted (in the patients home, hospital…)?
How many researchers read the data?
Please provide interview questions.
Results
Some tables need corrections, in Table 2 there is one Greek letter
None of the patients commented sexual life as issue, why?
You commented that health care system is important in the process of stoma adaptation and you mention community nurses and their role, at the same time in row 344 patient mention need for private care?
Author Response
- Introduction part:
I’d like to see more information in this part regarding overall according to age, gender and other sociodemographic characteristics
Reply: Thank you for the suggestion. We have added some relevant information about these in the Introduction part.
- Data collection:
I which town, area this study was conducted, is there any possibility for bias?
Reply: We have added this information in the section of Study Population and Sample.
- Why did you decide to interview 8 patients? Where this interview was conducted (in the patients home, hospital…)?
Reply: These points are described now in more detail in the section of Data Collection.
- How many researchers read the data?
Reply: We refer in that issue, in the section Trustworthiness of the research (lines 203-204: “Two researchers analyzed and interpreted the data so that the cross-referencing of the relevant data to lead to reliable results”).
- Please provide interview questions.
Reply: We have added the interview questions in the section of Data Collection.
- Results
Some tables need corrections, in Table 2 there is one Greek letter
Reply: Thank you for this remark. We have made the correction.
- None of the patients commented sexual life as issue, why?
Reply: Most of the patients referred to the problems encountered in their personal relationships, without referring directly to their sexual life. Maybe this is because people are hesitating to talk openly about their sexual life. There is, however, a passage in lines 270-275 that refer to this issue. Now we have described it in a clearer way.
- You commented that health care system is important in the process of stoma adaptation and you mention community nurses and their role, at the same time in row 344 patient mention need for private care?
Reply: Yes, this is true. Community nurses schedule their visits to patients on specific days and hours. When something urgent occurs to a patient at a time that there is not availability of community health care service (e.g. night hours) then the patient has to refer either to the nearest a tertiary hospital or to a private health care service.We explained this in the text more explicitly.

Reviewer 2 Report
I find the article good and useful. I recommend to the authors to add questions for the semi-structured interviews (in section Data collection) and recommendations for community nurses such as what shall they do foremost for their patients with permanent colostomy, what they should learn and in which areas they should develop (in section Conclusions).
There is a special symbol in Table 2 - probably a Greek letter instead of the letter C.
Author Response
- I recommend to the authors to add questions for the semi-structured interviews (in section Data collection) and recommendations for community nurses such as what shall they do foremost for their patients with permanent colostomy, what they should learn and in which areas they should develop (in section Conclusions).
Reply: We have added the questions used for the semi-structured interviews in the section of Data Collection. We have also made recommendations for community nurses in the Conclusions section.
- There is a special symbol in Table 2 - probably a Greek letter instead of the letter C.
Reply: Thank you for this remark. We have made the correction.

Reviewer 3 Report
Thank you for sending your paper entitled ““Living with a stoma”: Exploring the lived experience of patients with permanent colostomy.” to International Journal of Environmental Research and Public Health. After carefully review this interesting paper, the following comments are listed for your reference:
- Authors should ensure a thorough English and grammatical edit across the full manuscript. If sentences are not constructed in a grammatically correct manner it can impact meaning in unintended ways.To increase potential citations, authors should check keywords against those recommended in the MESH Browser of Medical Subject Headings https://meshb.nlm.nih.gov/search. On the other hand, please check the text for double spacing between words; there are too many to be listed.
- Methods (P3, L104-122): I would suggest to include the setting, as you do not mention where the study was conducted.
- Methods (P3, L124-131): It would be desirable to include the interview protocol, for example, in a table, to provide more context.
- Methods (P3-4, L141-154): Were the interviews conducted in English? If not, I would suggest explaining how the translation process was carried out in this section.
- Methods (P4, L141-154): Was the data from the interviews analysed using any software? If this is the case, it would be desirable to include this information in this section.
- Methods (P4, L156-166): Was this study approved by any Ethics Committee? If that is the case, I would recommend including the approval. Was the Declaration of Helsinki taken into account? If this is the case, I would suggest to include this information.
- Results (P5, L 200): In Table 2, I would suggest adding a column or row with a representative quote for each sub-theme that has emerged, as an example.
- Results (P5-8, L201-353): I would recommend keeping the same style with the quotes shown throughout the text and avoiding some quotes in cursive and others without it, to make it easier for the readers.
- Limitations (P11, L477-484): I would recommend to include in limitations section the fact that these findings cannot be extrapolated to wider populations, which is the main limitation of qualitative research.
- Conclusion: What clinical implications might this research have? What does this paper add to the field?
Author Response
- Authors should ensure a thorough English and grammatical edit across the full manuscript. If sentences are not constructed in a grammatically correct manner it can impact meaning in unintended ways.To increase potential citations, authors should check keywords against those recommended in the MESH Browser of Medical Subject Headings https://meshb.nlm.nih.gov/search. On the other hand, please check the text for double spacing between words; there are too many to be listed.
Reply: We have checked English grammar. We have also advised MESH Browser for keywords and we modified accordingly.
- Methods (P3, L104-122): I would suggest to include the setting, as you do not mention where the study was conducted.
Reply: We have added this information in the section of Study Population and Sample.
- Methods (P3, L124-131): It would be desirable to include the interview protocol, for example, in a table, to provide more context.
Reply: We have added the interview questions in the section of Data Collection.
- Methods (P3-4, L141-154): Were the interviews conducted in English? If not, I would suggest explaining how the translation process was carried out in this section.
Reply: Interviews were conducted in Greek. The participants’ passages were translated to English and accuracy of the translation was ensured by using a backward translation technique. This was added in the section of Data Collection.
- Methods (P4, L141-154): Was the data from the interviews analysed using any software? If this is the case, it would be desirable to include this information in this section.
Reply: The data analysis in the present study was performed manually. We have added this in the relevant section as per suggestion.
- Methods (P4, L156-166): Was this study approved by any Ethics Committee? If that is the case, I would recommend including the approval. Was the Declaration of Helsinki taken into account? If this is the case, I would suggest to include this information.
Reply: We have added relevant statements in the Ethics section.
- Results (P5, L 200): In Table 2, I would suggest adding a column or row with a representative quote for each sub-theme that has emerged, as an example.
Reply: We have added this as per suggestion.
- Results (P5-8, L201-353): I would recommend keeping the same style with the quotes shown throughout the text and avoiding some quotes in cursive and others without it, to make it easier for the readers.
Reply: We have modified the Results section as per suggestion.
- Limitations (P11, L477-484): I would recommend to include in limitations section the fact that these findings cannot be extrapolated to wider populations, which is the main limitation of qualitative research.
Reply: We have added this in the section of Limitations.
- Conclusion: What clinical implications might this research have? What does this paper add to the field?
Reply: We have added relevant information in the Conclusions as per suggestion.

Round 2
Reviewer 1 Report
-
Author Response
Thank you for your review report. I understand there are no comments.